# SARS-CoV-2 Testing of Aircraft Wastewater Shows That Mandatory Tests and Vaccination Pass before Boarding Did Not Prevent Massive Importation of Omicron Variant into Europe

**DOI:** 10.3390/v14071511

**Published:** 2022-07-09

**Authors:** Lorlane Le Targa, Nathalie Wurtz, Alexandre Lacoste, Gwilherm Penant, Priscilla Jardot, Alexandre Annessi, Philippe Colson, Bernard La Scola, Sarah Aherfi

**Affiliations:** 1Microbes Evolution PHylogénie et Infections, Institut de Recherche pour le Développement (IRD), Assistance Publique-Hôpitaux de Marseille (AP-HM), Aix-Marseille Université, 13005 Marseille, France; lorlane.le-targa@biosellal.com (L.L.T.); nathalie.wurtz@univ-amu.fr (N.W.); gwilherm.penant@ap-hm.fr (G.P.); priscilla.jardot@univ-amu.fr (P.J.); philippe.colson@univ-amu.fr (P.C.); 2Institut Hospitalo-Universitaire Méditerranée Infection, 19-21 Boulevard Jean Moulin, 13005 Marseille, France; 3Biosellal, 27 Chemin des Peupliers, 69570 Lyon, France; 4Bataillon des Marins Pompiers de la ville de Marseille, 13005 Marseille, France; alexandre.lacoste@bmpm.gouv.fr (A.L.); alexandre.annessi@bmpm.gouv.fr (A.A.)

**Keywords:** SARS-CoV2, wastewater monitoring, variant spread, aircraft, surveillance

## Abstract

Background: Most new SARS-CoV-2 epidemics in France occurred following the importation from abroad of emerging viral variants. Currently, the risk of new variants being imported is controlled based on a negative screening test (PCR or antigenic) and proof of up-to-date vaccine status, such as the International Air Transport Association travel pass. Methods: The wastewater from two planes arriving in Marseille (France) from Addis Ababa (Ethiopia) in December 2021 was tested by RT-PCR to detect SARS-CoV2 and screen for variants. These tests were carried out between landing and customs clearance and were then sequenced by MiSeq Illumina. Antigenic tests and sequencing by NovaSeq were carried out on respiratory samples collected from the 56 passengers on the second flight. Results: SARS-CoV-2 RNA suspected of being from the Omicron BA.1 variant was detected in the aircraft’s wastewater. SARS-CoV2 RNA was detected in 11 [20%) passengers and the Omicron BA.1 variant was identified. Conclusion: Our work shows the efficiency of aircraft wastewater testing to detect SARS-CoV-2 cases among travellers and to identify the viral genotype. It also highlights the low efficacy of the current control strategy for flights entering France from outside Europe, which combines a requirement to produce a vaccine pass and proof of a negative test before boarding.

## 1. Introduction

Coronavirus disease 2019 (COVID-19], caused by the Severe Acute Respiratory Syndrome Coronavirus 2 (SARS-CoV-2], emerged in Wuhan (China) in December 2019. Since then, it has become a pandemic, with more than 336 million confirmed cases globally and 5.5 million deaths as of 20 January 2022. Most national SARS-CoV-2 epidemics that occurred successively or concurrently resulted from the importation from abroad of emerging viral variants [1]. Air travel and cruises have been associated with the spread of SARS-CoV-2, including the spread of new variants via infected passengers. Since the beginning of the COVID-19 pandemic, many countries and regions have imposed restrictions, including quarantine periods, entry bans, compulsory vaccination and travel restrictions. For information on the restrictions imposed in different countries, see the COVID-19 Travel Regulations Map developed by the International Air Transport Association (IATA), powered by Timatic. This tool provides real-time information on travel requirements according to itineraries anywhere in the world (https://www.iatatravelcentre.com/world.php; accessed on 14 June 2022). Previous studies have examined the effect of travel restrictions and travel-related measures imposed during the pandemic. Most of these studies focused on the initial phase of the spread of COVID-19 when the epidemic was concentrated in Wuhan (China) [2,3]. All these studies found that travel restrictions in the early part of the epidemic helped to delay the spread of COVID-19. Other studies found that the restrictions were insufficient to completely control the global spread [4,5]. With the emergence of new SARS-CoV-2 variants, many countries have reinforced border control measures, including pre-travel and post-travel screening tests, to avoid the importation of these variants. In recent months, new variants have spread worldwide including, most recently, the Omicron variant which was first described in South Africa and Botswana [6]. Its clinical manifestations are similar to those of other respiratory viral infections with a dry cough, fever, tiredness, myalgia and breathing difficulties [6], but can also include gastrointestinal symptoms such as diarrhoea, nausea, abdominal pain and vomiting in between two and ten per cent of cases [7]. High concentrations of SARS-CoV-2 RNA have been found in the stools of infected asymptomatic and symptomatic people [8] and it has been shown that the virus remained infectious [9]. Therefore, analysis of SARS-CoV-2 in wastewater appears to be an interesting approach for monitoring the disease burden within communities. Since the first report of the detection of SARS-CoV-2 in wastewater by Medema et al. in the Netherlands [10], detection and monitoring in wastewater samples have been reported in many countries [11,12,13,14]. A few studies have performed SARS-CoV-2 genome sequencing from sewage to identify viral genotypes circulating within a community and to study genetic diversity [14,15,16,17,18,19]. Some of these showed a match between variants found in clinical isolates during the same period, while others identified genotypes not yet reported in clinical samples.

Until now, only three studies revealed that SARS-CoV-2 RNA had been detected in wastewater from passenger aircraft [20,21,22] and this monitoring demonstrated a high positive predictive value for SARS-CoV-2 infection among passengers. One of these works reported the successful detection by genome sequencing of variants in aircraft wastewater [20]. In our study, we report the detection of SARS-CoV-2 variants in the wastewater of aircraft travelling from Addis Ababa (Ethiopia) to Marseille (France). Two methods, including full-length genome sequencing and real-time reverse transcription (RT)-PCR (qPCR), to detect different variants by using the Bio-T Kit^®^ FiveStar COVID-19 (Biosellal, Dardilly, France) were used. Following the detection of a high concentration of the Omicron variant in the wastewater of the first aircraft, a wastewater sample from a second flight was tested and passengers were offered a test upon disembarking, to assess whether the results correlated. This indicated the widespread importation of the Omicron variant into France from Africa. This confirms that monitoring aircraft wastewater provides precious public health information on the global spread of emerging SARS-CoV-2 variants and shows that production of a negative SARS-CoV-2 test before boarding is no guarantee that passengers are not carrying the virus.

## 2. Materials and Methods

### 2.1. Samples

A volume of 100 mL of aircraft wastewater samples from two flights arriving from Addis Ababa (Ethiopia) to Marseille (France) on 22 and 24 December (hereinafter referred to as flights “2212” and “2412”, respectively) (Appendix A) were collected from the aircraft on the airport tarmac by the Bataillon des Marins Pompiers de Marseille (BMPM) via a special extraction valve which was washed between each sample by soaking for 15 min in bleach then 15 min in clean water. The samples were then stored at 4 °C until arrival at the laboratory. Samples were first passed through a paper filter to remove large particles, then a volume of 30 mL of the filtrate was filtered on a Millex sterile syringe filter with a pore size of 5 µm (SLSV025LS, Merck Millipore, Burlington, MA, USA).

For the flight on 24 December 2021, all passengers were offered a nasopharyngeal swab to test for SARS-CoV-2, in line with a joint initiative of the regional prefecture and the regional health agency and in compliance with Decree No. 2020-551 of 12 May 2020 on the information systems (Article 11 of Law No. 2020-546 of 11 May 2020 extending the state of health emergency for people arriving from countries experiencing active circulation of the virus) [23]. These nasopharyngeal swabs were taken by the BMPM staff from 56 passengers and were tested using a rapid antigenic diagnosis test COVID-VIRO^®^, AAZ (Boulogne-Billancourt, France). All positive samples were transported to our laboratory at 4 °C for further RT-PCR testing and sequencing (see below).

### 2.2. Nucleic Acid Extraction

Prior to DNA/RNA extraction, 10 µL of Bio-T Kit^®^ FiveStar COVID-19 internal positive control (Biosellal, Dardilly, France) and 10 µL of magnetic silica were added to each wastewater sample. Nucleic acids from 1 mL of each wastewater sample were extracted using the eGENE-UP^®^ Lysis and RNA/DNA Purification (BioMérieux, Marcy l’Etoile, France) to obtain a volume of 100µL of eluate. Negative controls consisted of RNase Free water which was extracted following the same protocol.

For clinical samples, viral RNA was extracted from a volume of 200µL using the KingFisher Flex system (Thermo Fisher Scientific, Waltham, MA, USA) to obtain a volume of 80 µL of eluate.

### 2.3. RT-PCR Detection and Variant Screening

Direct screening of SARS-CoV-2 variants in wastewater was performed using an RT-PCR Quantstudio5 device (Thermo Fisher, France) and a combination of the Bio-T Kit FiveStar COVID-19 and the Bio-T kit “Environmental Δ & O” (Biosellal, Dardilly, France). In addition, screening of N Gene (primers and probes of the CDC: www.cdc.gov/eid, accessed on 14 June 2022) was performed by RT-PCR, allowing assessment of the viral load. SARS-CoV-2 positive lateral flow results were confirmed by RT-PCR, as previously described [24].

### 2.4. Sample Preparation for NGS Sequencing

The first RNA/DNA extract from the 2212 sample was used without pre-treatment for further RT-PCR. For the 2412 sample, 1 mL was freeze-dried and then rehydrated in 30 µL of water. The reverse transcription step was performed in duplicate using the SuperScript VILO cDNA synthesis kit (11754-250, Thermo Fisher, Waltham, MA, USA) according to the supplier’s recommendations in a final volume of 20 µL per reaction. The ARTIC v3 PCR (ARTIC nCoV-2019 V3 Panel and 500 rxn of IDT 10006788, Integrated DNA Technologies, Inc., Coralville, IA, USA) was carried out under the following conditions for one reaction for each of the pools 1 and 2: 2.5 µL of reaction mix 10 × 0.5 µL dNTP [10 mM), 0.4 µL of forward primer [100 nM), 0.125 µL of HotStart qDNA Polymerase (Qiagen 203205, Hilden, Germany), water PCR grade (qsp 25µL) and 2 µL of template. Eight replicates were made per extract and per pool. The eight replicates were then pooled (final volume of 200 μL) before purification on a NucleoFast 96-well plate (Macherey Nagel ref 743100.50, Hoerdt, France). The purification products were eluted in 30 μL of TE 1X and then placed on a 2% agarose gel (migration for 30 min, 100 V). The 400 base pair (bp) bands were cut out of the gel and purified according to the supplier protocol using the Monarch DNA Gel Extraction Kit (New England BioLabs, ref T1020L, Évry-Courcouronnes, France) with a final elution volume of 40 µL.

For passengers’ samples, cDNA was amplified using the Illumina COVIDSeq protocol, including a multiplex PCR protocol with ARTIC nCoV-2019 V3 Panel primers (Integrated DNA technologies) according to the ARTIC procedure (https://artic.network/; accessed on 14 June 2022).

### 2.5. NGS Sequencing

For wastewater samples, final purification products were sequenced using the paired-end strategy with the Nextera XT DNA sample preparation kit (Illumina Inc., San Diego, CA, USA). These samples were barcoded for mixing with other projects. Libraries were prepared following the Illumina protocol (Illumina Inc., San Diego, CA, USA). PCR amplification to complete tag adapters and introduce dual index barcodes was performed over 12 cycles followed by purification with 0.8 × AMPure XP beads (Beckman Coulter Inc., Fullerton, CA, USA). Libraries were then normalised on specific beads according to the Nextera XT protocol (Illumina Inc., San Diego, CA, USA) and were then pooled. Automated cluster generation and pairwise sequencing with dual-index reads were performed in 2 × 250 bp using the Miseq Reagent Kit (V2-500 cycles) (Illumina Inc., San Diego, CA, USA). We chose to sequence wastewater samples using a MiSeq Illumina instrument to avoid possible cross contaminations with other clinical samples received in our laboratory and routinely sequenced on the NovaSeq 6000 instrument.

Viral genomes from clinical samples were sequenced using a NovaSeq 6000 instrument (Illumina Inc., San Diego, CA, USA), as previously described [25].

### 2.6. Sequence Analysis

Reads of wastewater samples were analysed as previously described [15]. Briefly, the reads from pool1 and pool2 provided by the ARTIC procedure were mapped together against the Wuhan-Hu-1 SARS-CoV-2 isolate genome (GenBank accession number NC_045512.2] using the CLC genomics softwarev7.5 (Qiagen Digital Insights, Hilden, Germany) with the default parameters. Non-synonymous mutations present in more than 10% of the reads were taken into account. For each sample, non-synonymous mutations were individually compared with classifying mutations that matched with 225 SARS-CoV-2 variants and sub-variants that had been in circulation since the beginning of the pandemic, including those that were circulating at this time. We refer to the non-synonymous mutations mapped during the analysis as ‘mutation patterns’. When a mutation pattern occurred in more than one variant or sub-variant, all variants and sub-variants were added to the results (Appendix A).

For clinical samples, genome consensus sequences were generated using the CLC Genomics workbench v.7 by mapping on the SARS-CoV-2 genome GenBank accession No. NC_045512.2 with the following thresholds: 0.8 for coverage and 0.9 for similarity. Sequences from complete genomes and clade assignments were analysed using the Nextclade web tool (https://clades.nextstrain.org/ accessed on 14 June 2022) [26].

## 3. Results

The results of screening the wastewater from flight 2212 were positive with a cycle threshold value (Ct) of 31.2, 29.8 and 34.4 for systems targeting the E gene, N gene and the E484A mutation, respectively, with a viral load of 171,699 copies/mL. The Ct value of the internal controls was 27. Wastewater from flight 2412 was also positive with a Ct value of 32.7, 30.6 and 34.9 for systems targeting the E gene, N gene and the E484A mutation, respectively, with a viral load of 95,846 copies/mL. The Ct value of the internal controls was 27.

For the 2212 sample, RNA concentration after extraction was 21.3 ng/µL. A total of 1,419,298 reads were obtained, and 97.3% were mapped, covering 75% of the reference genome. When considering a threshold of 10%, 31 non-synonymous mutations were present, 14 of them being signature mutations of SARS-CoV-2 variants (Appendix A). Furthermore, P4715L and D614G, which are present in the majority of SARS-CoV-2 variants, were found in 100% of the reads. Of 27 cumulative Omicron BA.1/21 K and BA.1.1 subvariant mutations covered by the reads, 13 were present at a frequency ranging from 51% to 100%. The following specific mutations of these subvariants were found: K856R, S3673_G3676S, I3758V, T547K and N856K. Of the 20 and 32 mutation patterns of the Omicron BA.1.1.529 and BA.2/21L subvariants covered by the reads, eight were found at a frequency ranging from 17% to 100%.

A total of 1,274,982 reads were obtained for the 2412 sample, of which 83.7% were mapped, covering 71.6% of the reference genome. Twenty-four non-synonymous mutations were present when considering a threshold of 10%. The analysis of the reads revealed 15 non-synonymous mutation patterns specific to variants (Appendix A). The D614G mutation was found in 100% of the reads. Of the 34 Omicron BA.1 and BA.1.1 mutation patterns covered by the reads, nine were present, with a frequency of between 36% and 100%. The K856R, S2083_L2084delinsIle, S3673_G3676S and T547K mutations specific to these sub-variants were found. Of a total of 27 and 39 mutation patterns of Omicron BA.1.1.529 and BA.2, the sub-variants covered by the reads, six were present, with a frequency ranging from 25% to 100%. Six mutation patterns specific to the Delta variant (comprising between 10 and 18 of the mapped mutations) were present, with a frequency ranging from 11% to 100%.

For the aircraft that arrived on 24 December 2021, of the 56 passengers who were administered a rapid antigenic diagnostic test upon disembarking the plane, 12 [21%) were detected as positive. The result was obtained within 20 min and was communicated directly to the passengers. Eleven of these positive tests were confirmed by RT-PCR, the results of Ct for each patient are summarized in Appendix A. SARS-CoV-2 next-generation genome sequencing using the COVIDSeq protocol was performed on these 11 samples, all of which were identified as the Omicron BA.1 variant.

## 4. Discussion

In this study, we screened the aircraft wastewater from two flights from Ethiopia to France. RT-PCR screening revealed the presence of the Omicron variant. Combined ARTIC and Illumina sequencing revealed the presence of mutation patterns of the Omicron variant. Interestingly, 12 passengers on the flight of 24 December 2021 tested positive using a lateral flow test and full genome sequencing of 11 of them revealed the presence of the Omicron variant. This number may also be underestimated since antigen tests are a little less sensitive than PCR tests. Thus, despite the requirement to have a negative PCR test within 72 h of boarding, as required even for vaccinated French nationals, one-fifth of the passengers on board this flight were infected. Furthermore, the congruent results of qPCR screening and variant detection by NGS of the aircraft wastewater and the genotype obtained on clinical samples showed that aircraft wastewater monitoring by NGS is effective for monitoring the circulation of variants, and is potentially a possible powerful strategy for preventing the widespread importation of new variants of concern from abroad. Of course, it would be important in future work to carry out an interrogation of the passengers to find out who went to the toilet during the trip, as well as the viral load of the positives, to determine as much as possible about the sensitivity of the procedure. However, since it is unthinkable to test all passengers on arrival, the detection of SARS-CoV-2 in the aircraft’s wastewater is still necessarily proof of the presence of infected people.

Monitoring SARS-CoV-2 circulation in wastewater has already proven to be an effective tool for tracking infections at the community level and has been correlated with the number of individual cases [27]. Applying such an approach to aircraft wastewater may be a powerful tool for controlling SARS-CoV-2 importation and exportation, a risk which exists despite strict measures to control passengers through mandatory clinical negative testing. Recent studies have shown that SARS-CoV-2 monitoring of wastewater from international flights and cruise ships is a useful way of prioritizing testing of passengers, and of improving the management of contact tracing [20,22]. Ahmed et al. recently detected the Omicron variant through NGS of aircraft wastewater samples collected from a flight arriving in Darwin (Northern territory, Australia) from Johannesburg (Gauteng, South Africa) [23]. They combined the ARTIC approach with Oxford Nanopore-GridION technology and ATOPlex combined with the DNBseq-g400 sequencing. In this study, we also detected the Omicron variant in wastewater samples from a long-haul (nine hours) flight from Addis Ababa to Marseille. Although the wastewater tanks were cleaned, drained, washed with detergent and drained again between flights, the possible contamination by remaining traces of SARS-CoV-2 RNA in the blackwater tanks from previous flights cannot be excluded. However, the Ct obtained by testing wastewater has to be taken into account. Here, the Ct were 27.99 and 30.24 (respectively for 2212 and 2412 flights) for the E gene corresponding to a viral load of 790,503 copies/mL and 169,462 copies/mL, respectively. This high viral load cannot be the result of remaining SARS-CoV2 RNA after cleaning tanks and is clearly linked to a viral excretion by onboard passengers, or even possibly the aircraft crew that was not tested for SARS-CoV-2 infection. The Omicron variant was designated as a variant of concern by the World Health Organization on 26 November 2021 and is described as highly transmissible, with a potential for immune escape as assessed by the reduced efficiency of the protective immunity developed after COVID-19 vaccination. Within three weeks of the first declared cases in Botswana, the Omicron variant had been detected in 87 countries [28]. Similarly, a previous study carried out in our laboratory showed that of sixteen SARS-CoV-2 variants identified in Marseille since the beginning of pandemic, seven had been imported through travel from abroad [1]. In order to learn from this experience, identify any potential pandemic waves from new emerging variants at an early stage and implement preventive measures, it has become indispensable to track the circulation of variants. The importation to Europe of the Omicron variant from South Africa illustrates perfectly that air traffic is a threatening and powerful entry point for new variants, despite public health policies including the “green” passport and compulsory RT-PCR testing for travelers (wrong certificates). This study shows that despite a requirement to show a negative RT-PCR test result within the 72 h before crossing the border, 20% of passengers were positive, highlighting a major failure of such prevention policies. Aircraft wastewater screening can be performed within an hour, a timeframe which is short enough to inform passengers before they clear customs, and for initiating nasopharyngeal testing and strict quarantine until the results are available. Such an approach could be an effective tool and is likely to be more powerful than presenting evidence of a negative test, which can be falsified. Aircraft and cruise wastewater monitoring may be of particular interest during inter-epidemic periods and in remote areas, where the massive importation of new SARS-CoV2 variants would have considerable public health consequences. Based on these findings, we propose SARS-CoV2 screening of wastewater followed by variant monitoring by NGS as a global strategy for preventing the importation of new SARS-CoV-2 variants to unaffected regions, especially in isolated areas such as islands or during periods of low viral circulation. We hope that the publication of such a manuscript on the subject of aircraft wastewater testing will allow us to convince our health authorities to carry out these tests on a large scale and not on an ad hoc basis but on an experimental basis, as the systematic screening and NGS of aircraft and boat wastewater may help policymakers to target management strategies by testing and isolating passengers in the event of the positive detection of SARS-CoV-2 in wastewater.

## Data Availability

Not applicable.

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
