# Peer review of "SARS-CoV-2 Testing of Aircraft Wastewater Shows That Mandatory Tests and Vaccination Pass before Boarding Did Not Prevent Massive Importation of Omicron Variant into Europe"

_viruses, 2022, doi:10.3390/v14071511_

Round 1
Reviewer 1 Report
I would like you to place a graph or table of the most important results
Author Response
We thank the reviewers for time and effort they have dedicated to providing their valuable feedback on our manuscript. We incorporated changes requested by the reviewers, in the aim to reply to all their comments and suggestions. Please find here a response to the reviewer’s comments and a new version of the manuscript with tracked changes to highlight the modifications done.
Please see the attachment.

Reviewer 2 Report
The present study reports on the RT-PCR analysis of wastewater from two aircraft. Also, the authors performed antigenic tests and sequencing on respiratory samples collected from the 56 passengers on the second flight. SARS-CoV-2 RNA suspected of being from the Omicron BA.1 variant was detected in the wastewater. In addition, SARS-CoV2 RNA was detected in 11 (20%) passengers, and the Omicron BA.1 variant was identified. The authors conclude that wastewater surveillance is an efficient method for detecting SARS-CoV-2 cases among travelers and identifying the viral genotype, highlighting the low efficacy of the current control strategies that combine a requirement to produce a vaccine pass and proof of a negative test before boarding.
The author’s initiative to investigate the potential role of wastewater surveillance is commendable. However, I have some major concerns regarding the strength of the present study:
1. Introduction: "Until now, only one study conducted early in the pandemic revealed that SARS-CoV-2 RNA had been detected in wastewater from passenger aircraft [20,21], and this monitoring demonstrated a high positive predictive value for SARS-CoV-2 infection among passengers [22]" The authors mention "one study," but they cite three studies.
2. Methods: The authors should provide more details on the wastewater samples: how were they collected?
3. What was the rationale for collecting 100 ml of wastewater? The aircraft wastewater is more concentrated than samples from community wastewater.
4. What measures were taken in order to avoid wastewater contamination? The lavatory wastewater chambers are cleaned after landing. Nonetheless, it is possible that traces of SARS-CoV-2 may remain in the wastewater chambers and may yield false-positive wastewater results for aircraft with no SARS-CoV-2 infection on board.
5. The authors used as a screening tool for passengers a rapid antigenic test. Therefore, they may have missed some positive cases.
6. Reporting the Ct values and the viral loads is essential. However, the authors did not provide any data on passengers' Cts/ viral load.
The rationale for testing the wastewater is to detect passengers with SARS-CoV-2 infection. However, this is of no use if the passengers are not infectious.
Therefore, a high-quality study must report if the passengers present infectious virus (documented by viral cultures) or at least serial Cts. The significance of positive results with high Ct is challenging to interpret without clinical history and context. Positive results with low viral load (high Ct) can be seen in the early stages of infection (before the person becomes capable of transmission of the infection) or late in infection when the risk of transmission is considered low. A swab taken at a single point in time does not provide information about the trajectory or subsequent course of illness. The serial trend of Ct values, which is linked to the probability of culturing live viruses, can predict likely individual infectiousness.
The WGS alone cannot prove the presence of infectious materials, as it does not require the presence of a viable virus.
If the passengers were not infectious, the positive wastewater samples are not of much use.
As in the present manuscript, a binary PCR result (positive/negative) does not tell us anything about the sensitivity of the screening test.
7. The authors detected the Omicron variant in wastewater samples and passengers. However, this does not narrow uncertainty on transmission (from passengers to wastewater). High-quality research would need to prove by genome sequencing and phylogenetic analysis that the samples from wastewater are from the presumed index cases, excluding contamination or co-infection.
8. The paraclinical data also should be corroborated with a thorough epidemiological investigation. The present study does not present any epidemiological data, which should be mandatory in studies investigating transmission. How many of the 56 passengers used the lavatory? Among them, how many RT-PCR-positive passengers used the lavatory?
9. In the absence of clinical testing results for onboard passengers, it is impossible to establish a link between the wastewater surveillance data and SARS-Cov-2 infected passengers. Such information is critical for determining the usefulness of wastewater surveillance as a complementary tool to clinical testing.
10. Also, a larger study would be very useful in order to determine the sensitivity and specificity of wastewater testing for detecting passengers with SARS-CoV-2 in aircraft.
Author Response
We thank the reviewers for time and effort they have dedicated to providing their valuable feedback on our manuscript. We incorporated changes requested by the reviewers, in the aim to reply to all their comments and suggestions. Please find here a point-by-point response to the reviewer’s comments and a new version of the manuscript with tracked changes to highlight the modifications done.
Please see the attachment.

Round 2
Reviewer 2 Report
The authors correctly addressed my previous comments.